# Molecular Mechanism behind the Safe Immunostimulatory Effect of *Withania somnifera*

**DOI:** 10.3390/biom13050828

**Published:** 2023-05-12

**Authors:** Kriti Kalpana, Shen Yap, Moriya Tsuji, Akira Kawamura

**Affiliations:** 1Biochemistry Ph.D. Program, The Graduate Center of CUNY, New York, NY 10016, USA; 2Department of Chemistry, Hunter College of CUNY, New York, NY 10065, USA; 3Aaron Diamond AIDS Research Center, Division of Infectious Diseases, Department of Medicine, Columbia University Irving Medical Center, New York, NY 10032, USA; 4Chemistry Ph.D. Program, The Graduate Center of CUNY, New York, NY 10016, USA

**Keywords:** endotoxin, inflammation, lipid A, MPLA, toll-like receptor 4, MYD88, TRIF, adjuvant, withaferin A, *Withania somnifera*

## Abstract

*Withania somnifera* (L.) Dunal (family *Solanaceae*) is a medicinal plant known for, among many pharmacological properties, an immune boosting effect. Our recent study revealed that its key immunostimulatory factor is lipopolysaccharide of plant-associated bacteria. This is peculiar, because, although LPS can elicit protective immunity, it is an extremely potent pro-inflammatory toxin (endotoxin). However, *W. somnifera* is not associated with such toxicity. In fact, despite the presence of LPS, it does not trigger massive inflammatory responses in macrophages. To gain insights into the safe immunostimulatory effect of *W. somnifera*, we conducted a mechanistic study on its major phytochemical constituent, withaferin A, which is known for anti-inflammatory activity. Endotoxin-triggered immunological responses in the presence and absence of withaferin A were characterized by both in vitro macrophage-based assay and in vivo cytokine profiling in mice. Collectively, our results demonstrate that withaferin A selectively attenuates the pro-inflammatory signaling triggered by endotoxin without impairing other immunological pathways. This finding provides a new conceptual framework to understand the safe immune-boosting effect of *W. somnifera* and possibly other medicinal plants. Furthermore, the finding opens a new opportunity to facilitate the development of safe immunotherapeutic agents, such as vaccine adjuvants.

## 1. Introduction

*Withania somnifera* (L.) Dunal (family *Solanaceae*), also known as Ashwagandha of Ayurvedic medicine, is a medicinal plant known for diverse pharmacological effects [1,2]. Among known effects of *W. somnifera* is its ability to stimulate the immune system [3,4,5,6]. Because of this immunostimulatory effect, *W. somnifera* has been examined for prevention or treatment of various infectious diseases, including listeriosis [7], DPT (diphtheria, pertussis, tetanus) [8], and COVID-19 [9]. Furthermore, *W. somnifera* has shown great promise as an immunological adjuvant for vaccines [4,8,10], which indicate the presence of a chemical factor that can stimulate antigen-presenting cells (APCs), such as macrophages and dendritic cells. However, the APC-stimulatory factor in *W. somnifera* remained a mystery for a long time. 

Clues to solve this mystery came from studies on other immune-boosting herbal remedies, namely, the genus *Echinacea* (hereafter “*Echinacea*”) [11] and Juzen-taiho-to [12]. APC-stimulatory factors in these herbal remedies have long been the subject of intensive research. Many studies have demonstrated that phytochemicals in *Echinacea*, such as alkamides [13] and arabinogalactans [14,15,16,17], exhibit diverse immunomodulatory effects. Likewise, numerous immunomodulatory phytochemicals have been identified for Juzen-taiho-to, including terpenes, flavonoids, phthalides, coumarins, and aromatic acids [12]. However, it remains to be determined whether these phytochemicals play important roles in the activation of APCs. In fact, alkamides have been reported to suppress the function of murine dendritic cells [18]. There is also a report of arabinogalactan proteins exhibiting weak stimulation of nitrite and IL6 production in a murine alveolar macrophage culture [15], which is in contrast to the potent APC-stimulatory effect of *Echinacea*. The vast majority of phytochemicals in Juzen-taiho-to exhibit anti-inflammatory effects, although Juzen-taiho-to, as a whole mixture, potently activates monocytes and macrophages through induction of inflammation and other immunological responses [12,19]. As such, these phytochemicals do not fully account for the potent APC-stimulatory effects of the original herbal remedies. Recently, a completely different class of compounds have emerged as the APC-stimulatory factors of *Echinacea* and Juzen-taiho-to. A series of studies on *Echinacea* [20,21,22,23] and Juzen-taiho-to [24,25] revealed that their key APC-stimulatory factors are not of plant origin. Instead, they are lipopolysaccharides (LPS) of plant-associated Gram-negative bacteria. This finding is supported by several lines of experimental evidence. First, the APC-stimulatory effects of *Echinacea* and Juzen-taiho-to diminish substantially when they are treated with polymyxin B, which depletes LPS [21,22,24]. Second, their APC-stimulatory effects correlate with bacterial load [22,23,25]. Third, *Echinacea purpurea*, when cultivated in a germ-free environment, do not stimulate macrophage production of TNF-α [20]. 

The finding of LPS in *Echinacea* and Juzen-taiho-to opened a possibility that APC-stimulatory effect of many other medicinal plants, including *W. somnifera*, could also arise from bacterial LPSs. In fact, when *W. somnifera* extracts were treated with Detoxi-Gel™, which removes LPS [26,27], the resulting samples no longer exhibited the APC-stimulatory effect [28]. Thus, LPS is indeed additionally a key APC-stimulatory factor in *W. somnifera*. 

The presence of LPS in *W. somnifera* raises an important new question. LPS is an extremely potent proinflammatory toxin. LPS and its glycolipid moiety, diphosphoryl lipid A (DPL), are potent agonists of toll-like receptor 4 (TLR4) (Figure 1A). Ligation of TLR4 triggers two signaling pathways, namely, (1) the myeloid differentiation marker 88 (MYD88) pathway, which mediates pro-inflammatory signaling, and (2) the toll–IL-1 receptor (TIR) domain-containing adaptor-inducing interferon-β (TRIF) pathway, which is associated with protective immunity through the induction of type-I interferons (IFN). LPS and DPL are known as “endotoxins” because they disproportionately activate the pro-inflammatory MYD88 pathway over the TRIF pathway (so-called “MYD88-bias”), which can result in massive inflammatory responses and toxicity (Figure 1B). The presence of such a potent pro-inflammatory toxin in *W. somnifera* is at odds with its long-tested safety in Ayurvedic medicine. 

Our recent study indicates that *W. somnifera* stimulates APCs in a manner similar to monophosphoryl lipid A (MPL) [28], which is a detoxified analog of DPL (Figure 1B) [29,30]. Unlike DPL, MPL exhibits much more attenuated activation of the MYD88 pathway, while retaining the ability to activate the TRIF pathway (Figure 1C). This balanced TLR4 activation makes it possible for MPL to stimulate the immune system safely. MPL is clinically used as an immunological adjuvant for various vaccines [31,32]. *W. somnifera*, despite the presence of LPS, also elicits MPL-like balanced TLR4 activation (Figure 1C) [28]. The lack of MYD88 bias suggests an as-yet uncharacterized mechanism by which the proinflammatory toxicity of LPS is attenuated in this plant. 

Here, we hypothesize that *W. somnifera* elicits the MPL-like balanced TLR4 activation because the endotoxin-induced MYD88 signaling is selectively counteracted by anti-inflammatory phytochemicals in this plant. In fact, *W. somnifera* contains withaferin A, which is a steroidal lactone originally isolated from the leaves of the plant [33]. Withaferin A is widely known for its potent anti-inflammatory activity [34]. Withaferin A is known to inhibit IKKβ [35], which, in turn, prevents the activation of NF-κB, the key mediator of the MYD88 signaling. What remains to be clarified, however, is whether withaferin A also modulates the TRIF pathway. This is an important question because, if withaferin A selectively attenuates the LPS-induced MYD88 signaling while keeping the TRIF signaling intact, it would explain the balanced MPL-like TLR4 activation by this plant. To test this hypothesis, we first conducted a macrophage-based assay to determine the effects of withaferin A on both MYD88 and TRIF pathways. The findings from this cell-based assay were further followed up with in vivo cytokine profiling, which led to a new conceptual framework to understand the safe immunostimulatory effect of *W. somnifera*. 

## 2. Materials and Methods

### 2.1. Materials

Phorbol 12-myristate 13-acetate (PMA) and withaferin A (Catalog number 681535, Lot number 2934717, purity 98.27% (HPLC), C_28_H_38_O_6_) were purchased from Sigma-Aldrich (St. Louis, MO, USA) and used without further purification. Reagents and supplies for qPCR were purchased from ThermoFisher Scientific (Waltham, MA, USA). Unless specified otherwise, all other chemicals and reagents were obtained from Fisher Scientific (Waltham, MA, USA) and VWR (Radnor, PA, USA) and used without further purification. 

### 2.2. Cell Treatment and Lysis for RT-qPCR Analysis

The detailed protocol for cell treatment and lysis has been published previously [28]. Briefly, human monocytic THP-1 cells were plated in a 12-well plate at 200,000 cells/mL of RPMI-1640 media, to which 25 nM PMA was added to differentiate the cells to macrophage phenotype. Cells were incubated for 48 h at 37 °C and 5% CO_2_. After 48 h of incubation, the media containing PMA was discarded and 2 mL of fresh media was added to the wells. This was followed by a rest period of 24 h in the absence of PMA during which differentiated THP-1 cells adhere to the tissue culture plate. The differentiated THP-1 cells were treated with DMSO (vehicle control), DPL (positive control, 5 µg/mL), MPL (5 µg/mL), and various mixtures of DPL (5 µg/mL) and withaferin A (0.1, 0.4, 0.8, 1.0 µg/mL). After 4 h of treatment, cells were lysed using 350 µL TRK Lysis Buffer (Omega Bio-Tek, Norcross, GA, USA) containing 2% β-mercaptoethanol, transferred to Omega^®^ Homogenizer columns (Omega Bio-Tek, Norcross, GA, USA), and centrifuged for 2 min at maximum speed (approximately 13,000 rpm). The homogenized lysate was either stored at −80 °C or immediately processed for RNA purification. 

### 2.3. RT-qPCR Assay for the Detection of MyD88 and TRIF Pathways

RNA purification, cDNA synthesis, and qPCR on an Applied Biosystems 7500 Real-Time PCR system were carried out as described previously [28]. The qPCR experiments used pre-optimized assays for IL-6 (FAM, ThermoFisher Assay Id: Hs00985639_m1), CCL5 (FAM, ThermoFisher Assay Id: Hs00982282_m1), and GAPDH endogenous control (ThermoFisher Catalog Number: 4325792). The ΔΔC_T_ method was employed to quantify the differential expression of IL-6 and CCL5. The raw data were first normalized by the endogenous control (GAPDH) for individual samples. Subsequently, relative quantification values, i.e., fold changes from the DMSO control, were obtained by comparing the normalized data against the DMSO vehicle control.

### 2.4. In Vivo Cytokine Profiling

All mouse procedures were approved by the Institutional Animal Care and Use Committee (IACUC) at CUNY Hunter College (Assurance #: AK-Cytokine 8/23). All mouse experiments were carried out in strict accordance with the Policy on Humane Care and Use of Laboratory Animals of the United States Public Health Service. BALB/c mice were treated (i.p.) with the following samples: (i) DPL (50 µg/mouse, n = 3), (ii) MPL (50 µg/mouse, n = 3), (iii) a mixture of DPL (50 µg/mouse) and withaferin A (0.1 µg/mouse) (n = 3); (iv) DMSO (vehicle control, n = 3). The blood (0.05–0.1 mL per animal) was collected at 6 h after injection. After separating the sera, the level of cytokines and chemokines was determined by Luminex Mouse Cytokine 32-Plex Discovery Assay at Eve Technologies (Calgary, AB, Canada), which quantified the abundance (pg/mL) of 32 cytokines and chemokines: namely, Eotaxin, G-CSF, GM-CSF, IFNγ, IL-1a, IL-1B, IL-2, IL-3, IL-4, IL-5, IL-6, IL-7, IL-9, IL-10, IL-12 (p40), IL-12 (p70), IL-13, IL-15, IL-17, IP-10, KC, LIF, LIX, MCP-1, M-CSF, MIG, MIP-1a, MIP-1B, MIP-2, RANTES, TNFa, VEGF.

## 3. Results

Figure 2 summarizes the main question addressed in this study. While withaferin A is known to inhibit IKKβ [35], which mediates the MYD88 signaling (Figure 2), it is unknown whether it also modulates the TRIF pathway. The MPL-like TLR4 activation by *W. somnifera* supports our hypothesis that withaferin A selectively inhibits the MYD88 pathway. On the other hand, a previous study on withaferin A and JAK/STAT activation led to a very different postulate: withaferin A might block the MYD88-independent, TRIF-dependent pathway rather than the MYD88-dependent pathway during LPS-induced TLR4 signaling [36]. To clarify the effects of withaferin A on TLR4 signaling, we set out to examine the expression levels of MYD88- and TRIF-regulated mRNA transcripts in macrophages treated with withaferin A and DPL (an endotoxic TLR4 agonist). 

### 3.1. Withaferin a Selectively Inhibits Pro-Inflammatory Signaling in DPL-Activated Macrophages

PMA-differentiated THP-1 cells, which exhibit macrophage phenotype, were treated with DPL (5 µg/mL) together with different concentrations of withaferin A (0.1, 0.4, 0.8, 1.0 µg/mL). After 4 h of incubation, which ensured the activation of TRIF signaling from endocytosed TLR4 (Figure 2), cells were lysed and subjected to RT-qPCR analyses of interleukin 6 (IL-6) and CCL5 to quantify the activation of MYD88 and TRIF pathways, respectively. 

Figure 3 shows dose–response effects of withaferin A on IL-6 and CCL5 mRNA levels in DPL-stimulated macrophages. In the absence of withaferin A, DPL exhibited the prototypical MYD88 bias of endotoxin, in which IL-6 was induced approximately 1000-fold compared to the vehicle control (DMSO), whereas CCL5 was induced a little over 10-fold. Addition of 0.1 µg/mL of withaferin A, however, reduced IL-6 induction to ~100-fold from the DMSO control, while CCL5 induction decreased only slightly. The trend continued as the concentration of withaferin A was further increased; at higher concentrations (0.8 and 1.0 µg/mL) the IL-6 level dropped precipitously below the basal expression level of the DMSO control; on the other hand, the CCL5 level, although somewhat decreased, remained well above the basal expression level. Collectively, these results indicate that withaferin A selectively attenuates the MYD88 signaling in DPL-activated macrophages. 

While the observed effects of withaferin A were striking, this cell-based study was limited in scope because only two mRNA transcripts were examined. In order to capture a broader and more physiologically relevant view of the effects of withaferin A on TLR4 signaling, we moved on to a follow-up study using in vivo cytokine profiling. 

### 3.2. Withaferin a Selectively Attenuates Pro-Inflammatory Cytokine Responses in DPL-Treated Mice

Mice were treated with DPL (50 µg/mouse), MPL (50 µg/mouse), a mixture of DPL (50 µg/mouse) and withaferin A (0.1 µg/mouse), and DMSO (vehicle control). The serum samples at 6 h after injection were subjected to Luminex multiplex assays to capture the snapshots of 32 cytokine/chemokine proteins. 

Figure 4 presents the overview of cytokine profiles in a radar chart; the data point of each axis (i.e., each cytokine) shows fold-change from the DMSO control. As expected, MPL (blue dots) and DPL (red dots) are clearly separated at multiple MYD88-regulated cytokines, such as IL-6, IFNγ, TNF, GM-CSF, and VEGF, whereas they overlap at many TRIF-regulated cytokines, such as CCL5 (RANTES), G-CSF, and IP-10. Based on how the mixture of DPL and withaferin A (“DPL + WA”, pale green line) overlaps with MPL and DPL, cytokines can be classified roughly into four groups: namely, (A) MPL-like, (B) DPL-like, (C) Similar to both MPL and DPL, and (D) Others. The first group is the MPL-like cytokines, which are expressed at similar levels in DPL + WA and MPL (highlighted in blue boxes). Many of them are pro-inflammatory cytokines, such as KC, IL-6, and VEGF. The second group is the DPL-like cytokines whose expression levels are similar in DPL + WA and DPL (highlighted in red boxes). They include pro-inflammatory (IL-1β), anti-inflammatory (IL-10) and TRIF-regulated (MCP-1) cytokines. The third group of cytokines are those whose expression levels are similar in DPL + WA, MPL, and DPL (highlighted in purple boxes). They include several TRIF-regulated cytokines, such as G-CSF and CCL5 (RANTES). The remaining cytokines comprise the fourth group (Others). Figure 5 shows the bar graphs of representative cytokines from each group. 

Overall, in vivo cytokine profiling results further support the notion that withaferin A selectively attenuates pro-inflammatory responses triggered by DPL, while leaving the TRIF-regulated cytokines intact. There are, however, notable differences in the cytokine profiles of DPL + WA and MPL, suggesting that the presence of withaferin A does not turn DPL into MPL. Rather, DPL + WA appears to be its own immunostimulatory entity with attenuated pro-inflammatory effects. 

## 4. Discussion

Our in vitro and in vivo studies demonstrate that withaferin A selectively attenuates the pro-inflammatory signaling triggered by DPL while keeping other immunological responses intact. The in vitro study showed that withaferin A can potently inhibit the DPL-induced MYD88 signaling in macrophages, which was quantified by the mRNA expression of IL-6, whereas the TRIF signaling, as quantified by CCL5, largely remained intact. The in vivo cytokine profiling allowed us to capture broad views of immunological responses to MPL, DPL, and DPL + WA, which enabled us to characterize the distinct effects of withaferin A. In particular, the in vivo study revealed notable differences between MPL and DPL + WA as well as their similarity in terms of the attenuated pro-inflammatory responses. As such, DPL + WA is not a mere replication of MPL. Rather, the mixture exhibits unique immunostimulatory effects of its own right. 

Our current finding provides a new mechanistic basis to understand the safe immunostimulatory effects of *W. somnifera* (Ashwagandha) [3,4,5,6]. Although *W. somnifera* contains bacterial LPS as the main immunostimulatory factor [28], the LPS-induced pro-inflammatory signaling is likely to be counteracted by anti-inflammatory phytochemicals (AIPs), including withaferin A. As a result, *W. somnifera*, as a whole, exhibits safe immunostimulatory effects. What remains to be determined is the contributions of other AIPs in *W. somnifera*. In addition to withaferin A, *W. somnifera* contains many other withanolides, such as withanolide A and withanone [3] as well as alkaloids and saponins [37]. It is possible that other chemical constituents also play roles in the immunological effects of the whole *W. somnifera* extract, which exhibited MPL-like balanced TLR4 activation (See AS1 and AS2 in Figure 1C) [28]. Available data, however, allows us to roughly estimate the contribution of withaferin A in the whole *W. somnifera* extracts. The observed effects of AS1 and AS2 were examined at 250 μg/mL. If the withaferin A content in Ashwagandha formulation is around 0.092% as reported previously [38], 250 μg/mL of Ashwagandha samples should contain roughly 0.2 μg/mL of withaferin A, which happens to be within the concentration range where withaferin A exhibited selective attenuation of pro-inflammatory signaling (Figure 3). As such, although more study is needed to clearly define the roles of individual AIPs in *W. somnifera*, withaferin A alone might be able to explain the balanced TLR4 activation observed for AS1 and AS2. 

The mechanistic model of APC-stimulation by *W. somnifera* offers a new conceptual framework to understand the safe immunostimulatory effects of other herbal remedies like *Echinacea* and Juzen-taiho-to, both of which are known to contain LPS [20,21,22,23,24,25]. As noted in the introductory section, these herbal remedies are ripe with structurally diverse AIPs. Those AIPs are likely to attenuate the pro-inflammatory toxicity of LPS, thereby ensuring the safety of these herbal remedies. It is tempting to speculate that, throughout the long history of herbal medicine, humans may have selected, through trial and error, medicinal herbs that contain selective inhibitors of the pro-inflammatory signaling triggered by LPS. In other words, herbal remedies that have been traditionally used to boost immunity may be a great source of AIPs that can selectively attenuate proinflammatory effects of endotoxins and possibly other pathogen-associated molecular patterns (PAMPs). 

Our current finding expands the emerging concept of combining PAMPs with anti-inflammatory drugs to generate safe vaccine adjuvants. This is an important concept that could transform the way new vaccine adjuvants are developed from PAMPs. Elimination of pro-inflammatory toxicity has been the major hurdle of adjuvant development from PAMPs. Currently, the standard approach is to structurally modify PAMPs in the hope of eliminating their intrinsic pro-inflammatory toxicity while maintaining the beneficial effects for protective immunity. In fact, MPL, the first non-alum adjuvant approved for clinical usage [28], was derived from DPL of *Salmonella minnesota* LPS through structural modifications (Figure 1B) [29,30]. However, structural modifications can be time-consuming, and there is no guarantee that the resulting PAMP analogs exhibit desirable immunological effects. Although the success of MPL spurred numerous studies to discover new adjuvants through structural modifications of PAMPs, the vast majority of such efforts, with the notable exception of CpG 1018 [39], did not yield vaccine adjuvants suitable for clinical usage. Unlike PAMP structural modification, the mixing of a PAMP and an anti-inflammatory drug can be done quickly for immunological characterization. In addition, it is possible to tune the immunological effects of the mixture in a predictable manner by changing the amount of an anti-inflammatory drug as exemplified in the dose–response profile of the DPL + WA mixture (Figure 3). As such, the concept of detoxifying PAMPs with anti-inflammatory drugs could greatly facilitate the discovery and development of new vaccine adjuvants. Such an approach was pioneered by Esser-Kahn and co-workers, who used an NF-κB inhibitor to suppress the pro-inflammatory effect of CpG, a TLR9 agonist, to obtain promising adjuvant candidates [40,41]. There is, however, an important difference between TLR9 and TLR4. While most toll-like receptors regulate either MYD88 or TRIF pathways (TLR9 regulates MYD88), TLR4 is the only one that control both MYD88 and TRIF pathways [42]. As such, modulation of TLR4 with small molecules, if it is possible, would allow us to control a broader range of immunological responses. Our current finding opens a possibility to use AIPs to fine-tune the immunological effects of TLR4 agonists to rapidly generate safe immunotherapeutic agents. After all, humans may have been using such an approach, albeit unknowingly, to safely boost immune functions for thousands of years in the practice of herbal medicine. 

## Figures and Tables

**Figure 1 biomolecules-13-00828-f001:**
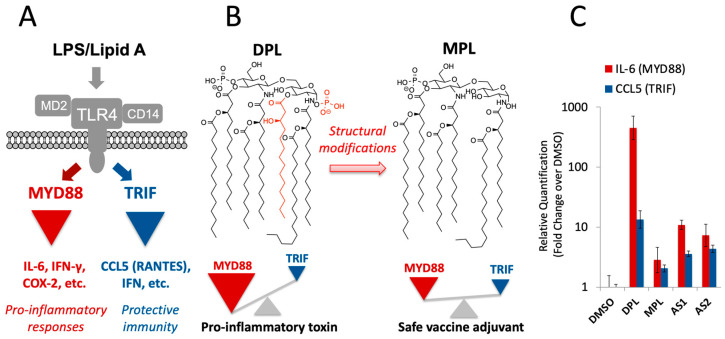
TLR4 signaling pathways. (**A**) Ligation of TLR4 with LPS/lipid A results in the activation of two pathways, namely, (1) pro-inflammatory MYD88 pathway and (2) TRIF pathway, which is associated with protective immunity. (**B**) Differential activation of TLR4 by DPL and MPL. DPL exhibits the MYD88 bias, which results in pro-inflammatory toxicity. On the other hand, MPL activates MYD88 and TRIF pathways more evenly, leading to safe stimulation of the immune system. Because of its ability to safely stimulate the immune system, MPL is used as an adjuvant for clinical vaccines. (**C**) *W. somnifera* exhibits MPL-like balanced TLR4 activation in macrophages. Two samples of *W. somnifera* (Ashwagandha), namely, “AS1” and “AS2,” exhibited balanced TLR4 activation profiles similar to that of MPL in an assay based on reverse transcription-quantitative polymerase chain reaction (RT-qPCR), in which the MYD88 and TRIF pathways are monitored by interleukin 6 (IL-6) and CCL5, respectively. DMSO (vehicle control); DPL (diphosphoryl lipid A of *E. coli*, endotoxin, 5 μg/mL); MPL (monophosphoryl lipid A, a clinical vaccine adjuvant, 5 μg/mL); AS1 and AS2 (Ashwagandha samples, 250 μg/mL). Each sample was analyzed in triplicate. Relative Quantification: fold change from the vehicle control (DMSO).

**Figure 2 biomolecules-13-00828-f002:**
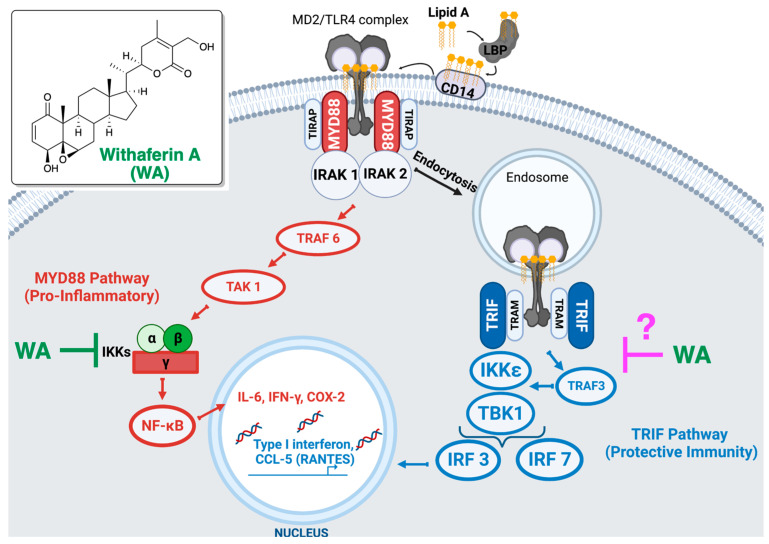
Possible effects of withaferin A on TLR4 signaling. Withaferin A is known to inhibit IKKβ, which mediates the MYD88 signaling. On the other hand, it is unknown if withaferin A modulates the TRIF pathway.

**Figure 3 biomolecules-13-00828-f003:**
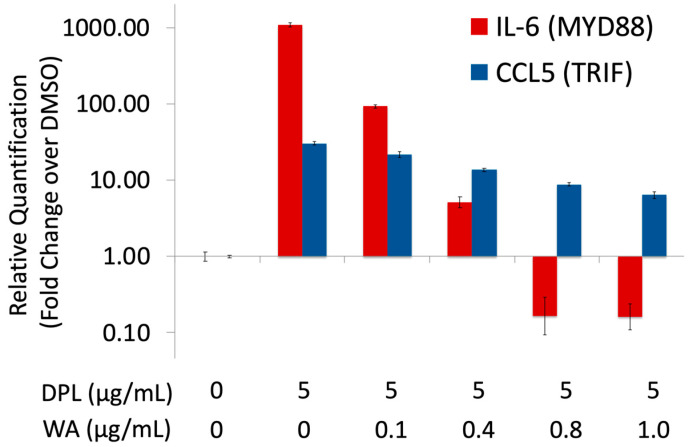
Effects of withaferin A on DPL-induced TLR4 signaling. PMA-differentiated THP-1 cells were treated with mixtures of DPL and withaferin A (WA) for 4 h and subjected to RT-qPCR assays of IL-6 (red) and CCL5 (blue) to quantify the activation of MYD88 and TRIF pathways, respectively. Except for the vehicle control (DMSO), the concentration of DPL was kept at 5 µg/mL, whereas WA concentration was varied from 0.1 to 1.0 µg/mL to examine its dose–response profile. Each sample was analyzed in triplicate. The y-axis is relative quantification, which is fold change from the vehicle control (DMSO).

**Figure 4 biomolecules-13-00828-f004:**
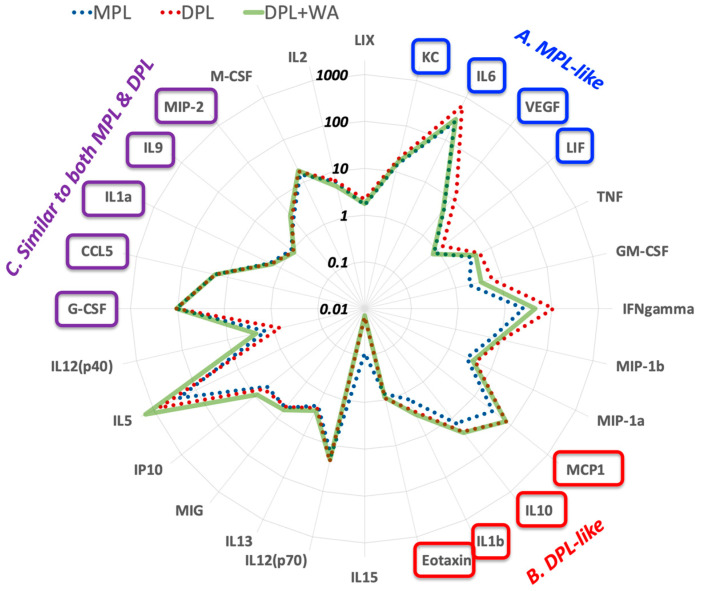
The radar chart of in vivo cytokine profiles in mice treated with MPL (blue dots), DPL (red dots), and DPL + WA (pale green line). Out of 32 cytokines/chemokines studied, four of them (IL-3, IL-4, IL-7, and IL-17) were removed from the chart due to low expression. A data point on each axis represents fold-change of the corresponding cytokine expression from the DMSO control. Cytokines are classfied into four groups based on how DPL + WA compares to MPL and DPL. (A) MPL-like cytokines (highlighted in blue boxes). (B) DPL-like cytokines (highlighted in red boxes). (C) Similar to both MPL and DPL (highlighted in purple boxes).

**Figure 5 biomolecules-13-00828-f005:**
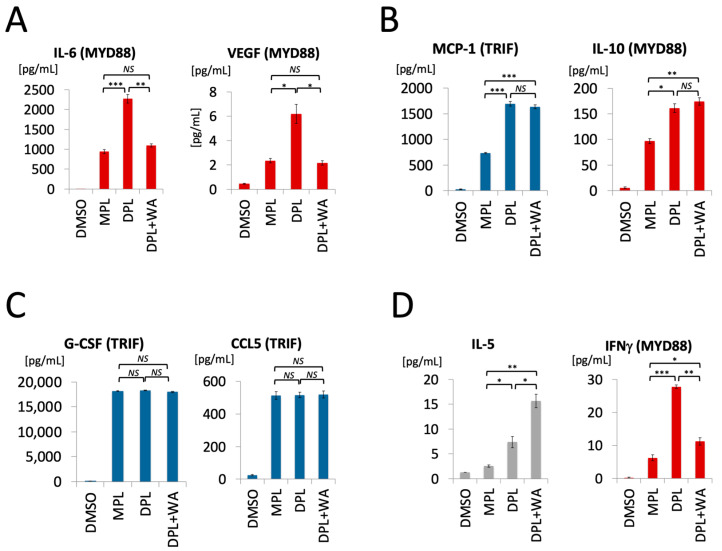
Bar graphs of representative cytokines in the four groups. Cytokines were classfied into four groups based on how DPL + WA compared to MPL and DPL (see Figure 4). (**A**) MPL-like cytokines. (**B**) DPL-like cytokines. (**C**) Similar to both MPL and DPL. (**D**) Others. * *p* < 0.05, ** *p* < 0.005, *** *p* < 0.001 (*t*-test), NS: not significant.

## Data Availability

The data presented in this study are available in the figures of this manuscript. Raw data can be made available upon request.

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
