# Peer review of "Molecular Mechanism behind the Safe Immunostimulatory Effect of Withania somnifera"

_biomolecules, 2023, doi:10.3390/biom13050828_

Round 1
Reviewer 1 Report
The authors have investigated the molecular mechanisms involved in the immunomodulatory effect of a well-known medicinal plant, Withania somnifera, using both in vitro and in vivo techniques. The manuscript brings important data useful to a wide audience interested in phytotherapy. However, a few aspects need to be better explained/clarified in the manuscript, ahead of publishing:
1. The idea that bacterial contaminants are responsible for the immunostimulant effects of certain plants is certainly interesting but still controversial. There are a lot of researchers that still believe that alkamides and arabinogalactans from Echinacea are responsible for the immunostimulant effect of the mentioned plant, and not bacterial compounds. The whole paragraph from Lines 38-46 suggests that surely it is the bacterial LPS from W. somnifera which is activating an immune response. Although it may be true (partially), other mechanisms could also be involved, which should have been mentioned in the manuscript's introduction.
2. Other steroidal lactones are present in the chemical composition of W. somnifera, apart from withaferin A, and they also may have a role in the immune effects of the plant. In fact, a previous article of the authors proved that the supplements AS1 and AS2 (with whole W. somnifera extracts) had relatively the same effects on THP-1 cells as withaferin A, used in the current study. Some comparisons and additional discussions pointing to other (possible) molecular candidates are necessary in the current manuscript.
3. Minor issue:
- at the first mentioning in the text of a plant species, plant Family should also be given
English language is fine.
Reviewer 2 Report
This study provides a new mechanistic basis to understand the safe immunostimulatory effects of W. somniferadetermining the effects of withaferin A on both MYD88 and TRIF pathways.
This work represents an interesting biochemical study not only in vitro but also in vivo which demonstrates that withaferin A selectively attenuates the pro-inflammatory signaling (LPS-induced MYD88) triggered by DPL while keeping other immunological responses (TRIF signaling) intact.
The authors used a clear methodology and explain the work in a very logical pathway formulating clear hypotheses and the way to reach their objectives, with clear illustrations which support very well their demonstration.
Therefore it deserves publication in Biomolecules.
However some minor points needed to be taken into account before the acceptance
L 26 could you revise the key words: the name of the plant and the name of whitaferin A will be appropriated
L 29 Withania somnifera–could you add for the first time the taxonomist (L.) Dunal
L91 “In fact, W. somnifera contains withaferin A, ……………”
Could you add here after whitaferin A : a whitanolide of the steroid lactone type and some words of the sources of this compound which has been originally extracted from the plant of Whitania somnifera , precising which part of the plant ?
L 106 could you add a sentence with some characteristics on the structure (CHO, molecular weight and absolutely the degree of purity of this compounds ? and the Hplc control? and the endotoxin free assay?
L 141 could you add the strain of mice?
43.1. Witherferin A --->Whitaferin A
could you check again all the possible spelling mistakes in the manuscript ?
References: all the Latin names in the references and also in the manuscript have to be italicized.
could you check the list of abbreviations? It seems that some are missing. As exemple AIPs should appear in the list. Could you check others?
Round 2
Reviewer 1 Report
The authors have improved their manuscript according to suggestions, therefore it can be published in the revised form.